# Impact of Hyperspectral Infrared Sounding Observation and Principal-Component-Score Assimilation on the Accuracy of High-Impact Weather Prediction

Qi Zhang [1,2] and Min Shao [1,*]

1 School of Environment, Nanjing Normal University, Nanjing 210023, China; qzhang487@wisc.edu
2 Space Science and Engineering Center, University of Wisconsin-Madison, Madison, WI 53706, USA
* Correspondence: mshao@masonlive.gmu.edu

**Abstract:** Observations from a hyperspectral infrared (IR) sounding interferometer such as the Infrared Atmospheric Sounding Interferometer (IASI) and the Cross-Track Infrared Sounder (CrIS) are crucial to numerical weather prediction (NWP). By measuring radiance at the top of the atmosphere using thousands of channels, these observations convey accurate atmospheric information to the initial condition through data assimilation (DA) schemes. The massive data volume has pushed the community to develop novel approaches to reduce the number of assimilated channels while retaining as much information content as possible. Thus, channel-selection schemes have become widely accepted in every NWP center. Two significant limitations of channel-selection schemes are (1) the deficiency in retaining the observational information content and (2) the higher cross-channel correlation in the observational error (R) matrix. This paper introduces a hyperspectral IR observation DA scheme in the principal component (PC) space. Four-month performance comparison case studies using the Weather Research and Forecasting model (WRF) as a forecast module between PC-score assimilation and the selected-channel assimilation experiment show that the PC-score assimilation scheme can reduce the initial condition's root-mean-squared error for temperature and water vapor compared to the channel-selection scheme and thus improve the forecasting of precipitation and high-impact weather. Case studies using the Unified Forecast System Short-Range Weather (UFS-SRW) application as forecast module also indicate that the positive impact can be retained among different NWP models.

**Keywords:** hyperspectral; infrared; principal component; data assimilation; weather prediction

## 1. Introduction

The use of hyperspectral infrared satellite sounding observations in data assimilation systems remarkably improves the accuracy of numerical weather prediction models [1,2]. Operational centers assimilate these observations in the channel space by selecting a sub-set of channels [3–7]. The channel-selection method brings a balance to NWP between the computational efficiency and information content but leaves two issues unsolved: (1) information content deficiency and (2) cross-channel error correlation. For issue (1), various experiments have successfully increased the assimilated information content by introducing more hyperspectral sounding channels into the DA system [8–10]. However, the information content from the selected channels remains lower than the original observation [11,12]. For issue (2), multiple studies have demonstrated the feasibility of alleviating cross-channel correlation in the observational error covariance (R) matrix by introducing a cross-channel correlation-aware module into channel-selection schemes [13–17]. However, these approaches can barely reach an equilibrium between adding more channels in the data assimilation process and the cross-channel correlation increase in the R matrix. This is because hyperspectral IR sounders reduce the measurement error by adding more correlated channels in the IR spectral region to increase the information redundancy. Both issues could

restrict DA–NWP systems from taking full advantage of the next-generation low Earth orbit hyperspectral IR sounders, e.g., Infrared Atmospheric Sounder Interferometer—New Generation (IASI-NG), because of the higher spectral resolution [18].

Principal component analysis (PCA) [19] provides a better solution to the above issues by converting the IR spectral observations to "imaginary" orthometric variables— principal component (PC) scores. Studies using various hyperspectral IR sounding instruments have revealed the capability of PCA to eliminate information redundancy, retaining crucial independent information content, and improving the signal-to-noise ratio [20–23]. Such advantages may indicate that the assimilation of PC scores can correct the initial condition better than a channel-selection scheme, resulting in more noise-reduced and independent observational information. Furthermore, the orthogonality of PCs can reduce cross-channel correlation in the R matrix (Figure A1 in Appendix A). PC-based fast radiative-transfer models such as the principal-component-based radiative-transfer model (PCRTM) [24], the Havemann–Taylor fast radiative transfer code (HTFRTC) [25], and principal-component-based radiative transfer for TOVS (PC-RTTOV) [26] possess higher computational efficiency than channel-based models, for example the community radiative-transfer model (CRTM) [27] and the radiative transfer for TOVS (RTTOV) [28]. In addition, a study conducted by Matricardi and McNally [29] found no detectable loss of accuracy and notable computational efficiency in the ECMWF's 4DVar DA system when assimilating 20 PC scores derived from the 165-channel radiance observations. Other investigations [30,31] have also confirmed the capability of the PC score to retain observational information via different approaches:

(1) Matricardi and McNally [29] generated 20 PC scores from 165 out of 8461 IASI channels in their research, which meant a genuine portion of observational information was unavoidably discarded. Moreover, the PC-based fast-forward radiative-transfer model (RTM) in their study can only assimilate IASI observations, meaning further validation from other experiments using different instruments.

(2) Collard et al. [30] focused on the noise cancellation capability of PCA and its impact on the DA system by assimilating reconstructed radiance observations via a channel-selection method. Their results indicated that signal-to-noise ratio improvement (noise cancellation) can enhance the impact of IR radiance observations in the DA system.

(3) Lu and Zhang [31] highlighted the PC scores' information content preservation capability, but the PC scores are still not generated from full-spectrum radiance observation.

Based on previous studies [29–31], three modifications were made in the present study. First, observed PC scores were derived from full-spectrum radiance observations. Second, the R matrix was generated from RTM-simulated and observation-derived PC scores to better represent the observational error. Third, a PC-based RTM was deployed to simulate multiple hyperspectral IR sounding observations. To specifically exploit hyperspectral IR observation DA's impact on NWP, no other remote-sensed observations were assimilated in the experiments; conventional observations were excluded from DA for independent performance evaluation.

Following the introduction, Section 2 focuses on the methods for generating the background and observation error covariance, the structure of cost function, and the NWP system workflow. Section 3 presents three case-study comparisons between using the PC score and channel-selection DA scheme relative to a baseline experiment which does not assimilate any observation, a 4-month-long forecast performance evaluation between PC-score assimilation and the channel-selection DA (CTL) experiment, and two case studies using the Unified Forecast System Short-Range Weather (UFS-SRW) application at a cloud-resolving horizontal resolution (3 km). The results are discussed in Section 4. Section 5 provides our conclusions.

## 2. Materials and Methods

Before starting the experiments, a hybrid 3-dimensional variational (3Dvar) DA system was constructed to assimilate hyperspectral IR sounding observations in PC and channel space [32]. The cost function in the system uses a hybrid 3DVar scheme [33], which is equivalent to the NOAA/NCEP Rapid Refresh (RAP) assimilation and model forecast system [34]. The DA system generates the analysis field by minimizing the cost function:

$$J_{(x)} = (x - x_b)^T B_{hyb}^{-1} (x - x_b) + \left( y - H_{(x)} \right)^T R^{-1} \left( y - H_{(x)} \right) \tag{1}$$

where

$$B_{hyb} = a \cdot B_{static} + b \cdot B_{ensemble} \tag{2}$$

The flow-dependent background error covariance $B_{hyb}$ in the hybrid 3Dvar combines the static background error covariance matrix $B_{static}$ and the ensemble background error covariance matrix $B_{ensemble}$. The $B_{static}$ is created using the NMC method [35] from a 6-month-long RAP model forecast dataset (from 1 March 2021 to 31 August 2021) [36], and the $B_{ensemble}$ is generated from a 31-member forecast output from the NOAA Global Ensemble Forecast System (GEFS). In the following experiments, the $B_{hyb}$ consisted of 80% $B_{static}$ and 20% $B_{ensemble}$. The observational part can be linearized to:

$$y - H_{(x_b)} - H'_{(x_b)} \cdot (x - x_b) \tag{3}$$

where $H$ is the radiative-transfer model and $H'$ is its Jacobian. To ensure the DA system's capability to perform radiance- and PC-based Jacobian calculation, we used HTFRTC as the radiative-transfer model and incorporated two DA approaches into the system. The system calculated the Jacobian in PC (radiance) space in the PC-score (selected-channel radiance) assimilation experiments.

Currently, the system can assimilate observations from IASI and CrIS. Their observational error covariances $R$ are calculated in the PC and channel space using the Hollingsworth–Lönnberg method [37,38] from observations gathered between 1 March 2021 and 31 August 2021. The data used in the calculation are available on the NOAA Comprehensive Large Array-Data Stewardship System. For channel-based DA, the matrices only include the channels adopted by ECMWF [4] and NCEP's [8] data assimilation system (Figure 1b,d) in the calculation. For PC-based DA, we first converted full-channel observations from IASI or CrIS to 30 PC scores using HTFRTC's PC coefficients then generated the matrices (Figure 1a,c). Comparing the PC-based $R$ matrices with the channel-based ones, the PC score's capability to diminish the cross-channel error correlation is superior to that of the channel-selection scheme. The selected-channel radiance and the weighting function to temperature and water vapor of the first 30 PCs (Figure A2 in Appendix A) have high similarity, which indicates that both assimilation methods have the capability to modify the tropospheric temperature and water vapor field.

Before initializing the DA process, cloud screening and quality control have to be conducted to ensure (1) the observations are from clear-sky regions and (2) the quality of each observation. In cloud detection, the clear-sky percentage in each field of view (FOV) is determined by the Advanced Baseline Imager (ABI) Full Disk Clear Sky Mask from GOES-16 (https://noaa-goes16.s3.amazonaws.com/index.html#ABI-L2-ACMF/, accessed on 5 February 2023): If the ABI clear sky pixel amount within an observational (IASI or CrIS) FOV is less than 80% of the total ABI pixel amount in the same FOV, then the observation is discarded. After screening out the cloudy observations, the quality control process excludes the disqualified clear-sky observation using a method described as follows: the quality control converts clear-sky radiance observations to PC scores via QR decomposition and calculates the first 30 PC scores' mean bias ($MB$) and standard deviation ($SD$) with reference to the simulated PC scores from the first guess; if more than 25 PC scores from an observation are located within $MB \pm 1.5SD$, then all 30 PC scores are assimilated by the DA system. The PC-score assimilation and radiance assimilation experiment shares

the same cloud-screening and quality control method, but for the radiance assimilation, the PC scores are converted back to spectral radiance after quality control. In this case, the radiance assimilation experiment shares the same approach mentioned in Collard et al.'s experiment [30].

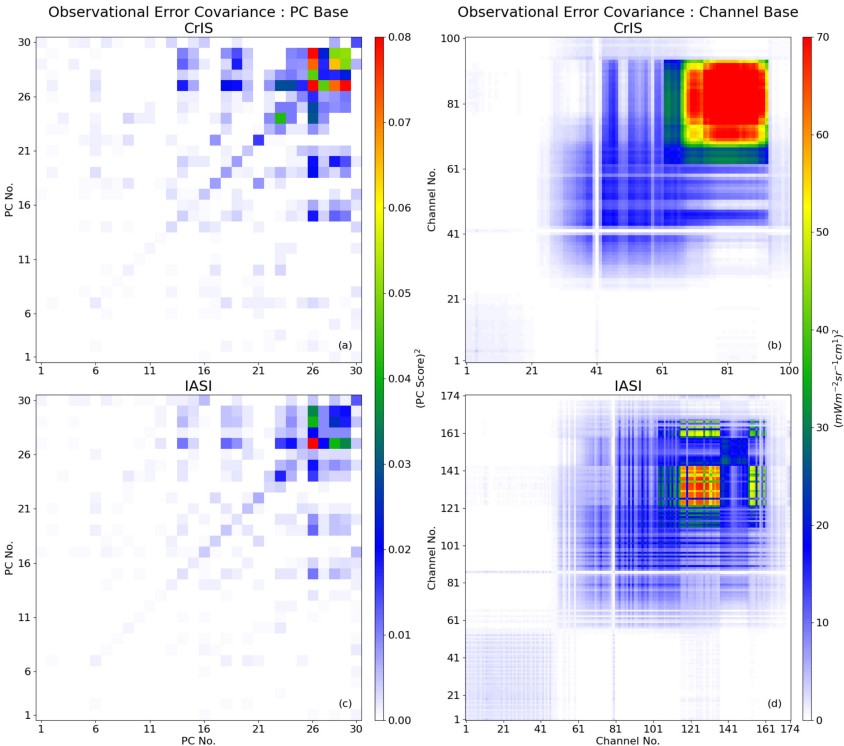

**Figure 1.** Observational error covariance in PC space (**a**,**c**) and channel space (**b**,**d**) for CrIS (**a**,**b**) and IASI (**c**,**d**).

The NWP model used in the experiment is the Advanced Research version of the WRF [39]. The forecast domain (Figure 2) covers the conterminous United States with a 13 km horizontal resolution. Table 1 lists the primary attributes of the WRF configuration in the following experiments. This configuration shares a high degree of similarity with the operational RAP system [36]. In the following experiments, the initial and boundary condition are provided by the RAP forecast product.

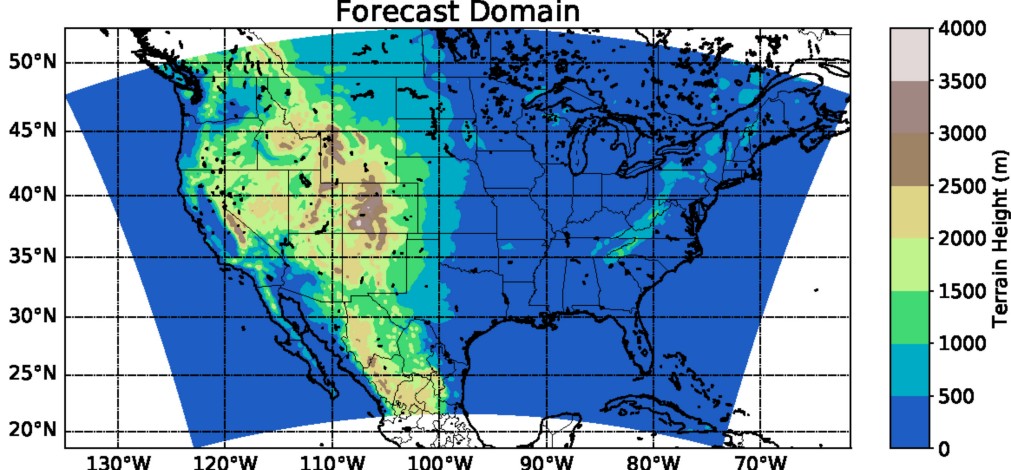

**Figure 2.** Spatial coverage of the forecast domain.

**Table 1.** Model and physics configuration.

| Model Settings | |
|---|---|
| Version | ARW 4.3, non-hydrostatic |
| Map projection | Lambert |
| Grid points | $400 \times 257$ |
| Vertical Layers | 51 |
| Model top | 50 |
| Lateral boundary conditions | RAP |
| Horizontal/Vertical Advection | Fifth-order upwind |
| Time step | Adjusted time step, maximum 45 s. |
| Damping option | Rayleigh, dampcoef = $0.2s^{-1}$, zdamp = 5000m |
| Horizontal diffusion | Sixth-order (0.12) |
| Forecast lead time | 18 h |
| Radiation scheme | RRTMG |
| Land surface scheme | RUC |
| Land use category | MODIS 24 category |
| Planetary-boudary and surface layer scheme | MYNN |
| Shallow convection scheme | Grell-Freitas |
| Deep convection scheme | Grell-Freitas |
| Cloud Microphysics scheme | Thompson aerosol aware |

In each experiment (Figure 3), the forecasts are initialized twice a day, at 00:00 and 12:00 UTC. The system finishes 12 analysis cycles before commencing the forecast cycle. Like RAP, the initial conditions for analysis cycles 2 to 12 derive from the former analysis cycle's 1 h lead-time forecast output. The exception is analysis cycle 1, the initial condition of which is inherited from the RAP analysis field. The DA process is triggered if the IASI or CrIS observation is received within the 1 h assimilation time window (plus and minus 30 min relative to the analysis validation time). For the analysis cycles' lack of available observations, a one-hour lead-time forecast is generated without launching the DA process. In general, there are 8 out of 12 analysis cycles which have assimilated observations.

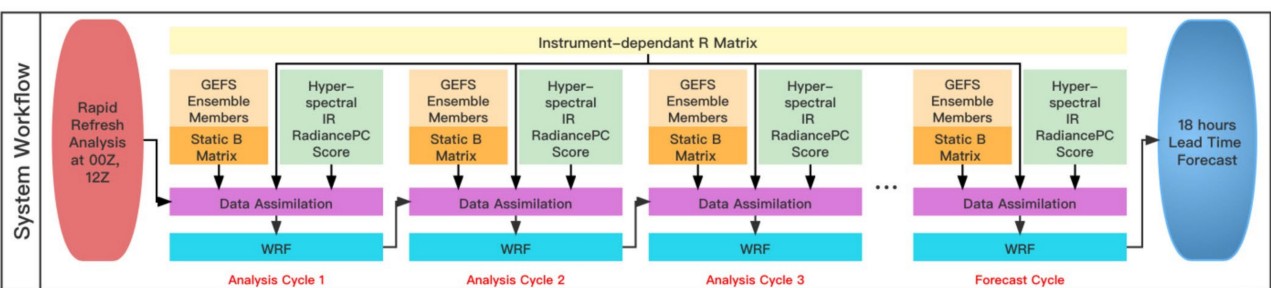

**Figure 3.** Forecast system workflow.

## 3. Results

### 3.1. PC-Score vs. Selected-Channel Radiance Assimilation: Case Studies

Three massive tornado outbreak cases were selected—2019Mar03 (12:00 UTC 3 March 2019 to 12:00 UTC 4 March 2019); 2020Mar03 (12:00 UTC 3 March 2020 to 12:00 UTC 4 March 2020); and 2020Apr12 (12:00 UTC 12 April 2020 to 12:00 UTC 13 April 2020)—to investigate the performance of PC-score assimilation relative to selected-channel radiance assimilation.

A control experiment was also conducted to summarize the baseline performance of the NWP system.

The root-mean-squared error (*RMSE*) can reveal the difference in model performance by measuring the departure between estimation and observation. Equation (4), where (*Model*) represents the foreast result from experiment (*OBS*) and *OBS Amount* is the number of observations, was used to derive the temperature and specific humidity RMSE profile from the initial condition's departure from aircraft and radiosonde observations. The observations were gathered from the NOAA/NCEP website (https://registry.opendata.aws/noaa-rap, accessed on 5 February 2022). After deducting the baseline experiment (forecast that does not assimilate any observation) RMSE from the PC-score assimilation and selected-channel radiance assimilation experiment, these phenomena can be clearly detected: the temperature in the initial condition of PC-score assimilation has a smaller RMSE, especially at levels from 900 to 700 and above 600 hPa (Figure 4a). This reduction in RMSE indicates the initial condition generated by PC-score assimilation (red line) is better than that of selected-channel radiance assimilation (green line). The negative specific humidity RMSE divergence above 800 hPa (Figure 4b) indicates that the water vapor field that comes from the PC-score assimilation experiment quantitatively shares the same performance as the temperature field when compared with the selected-channel radiance assimilation.

$$RMSE = \frac{\sqrt{\sum(Model - OBS)^2}}{OBS\ Amount} \tag{4}$$

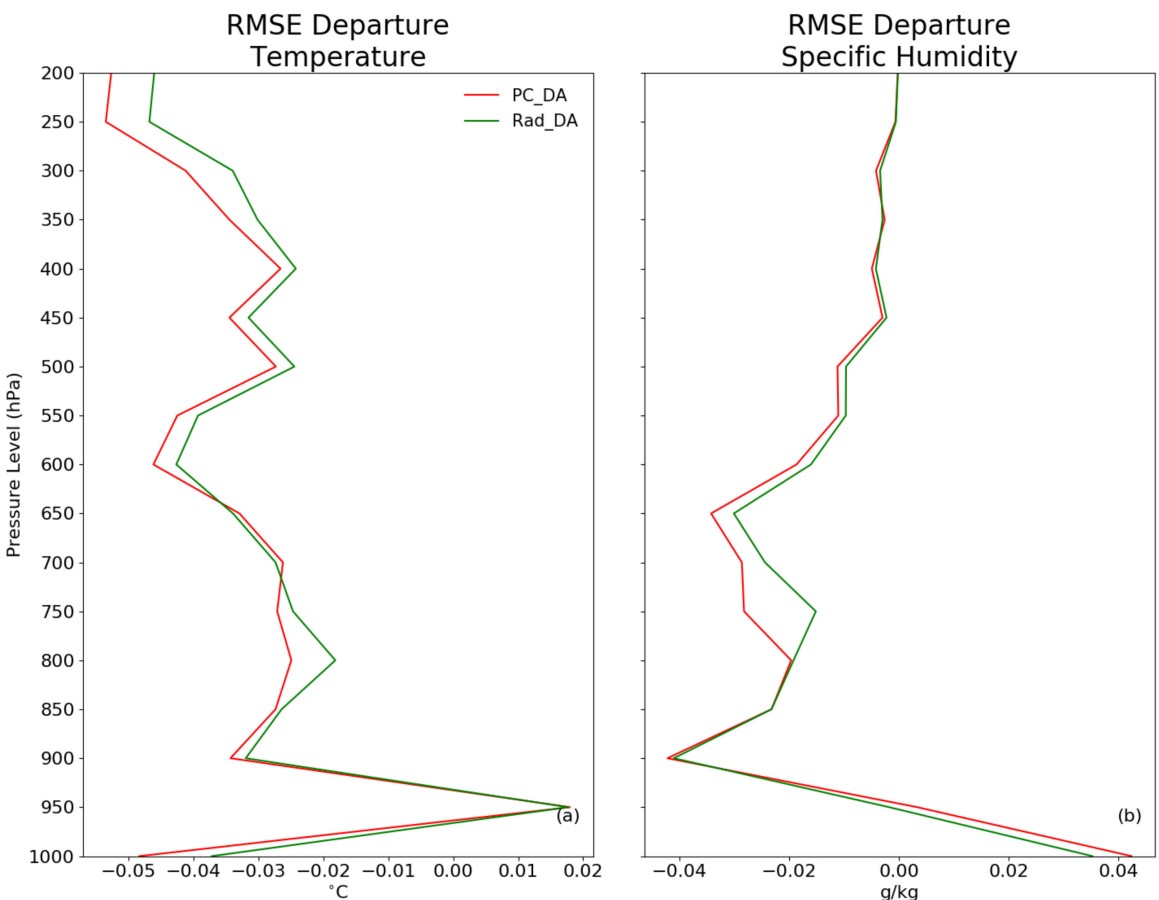

**Figure 4.** Temperature (**a**) and specific humidity (**b**) RMSE departure of PC-score assimilation (red line) and selected-channel radiance assimilation (green line).

Taking NCEP Stage IV quantitative precipitation estimation [40] as the unbiased observation, each experiment's critical success index ($CSI = \frac{a}{a+b+c}$) was calculated to evaluate the precipitation prediction ability. The contingency table (Table A1) in Appendix A explains the variables in the CSI calculation. Like the approach for evaluating temperature and water vapor, PC-score assimilation and selected-channel radiance assimilation remove the baseline CSI (derived from the control experiment) to make the difference more legible (Figure 5). For moderate precipitation, the PC-score assimilation experiment's positive CSI departure from the selected-channel assimilation experiment indicates it improves its ability to predict rainfall between 2.5 mm/h and 7.5 mm/h. For heavy precipitation (>7.5 mm/h), PC-score assimilation still has an advantage over selected-channel radiance assimilation, albeit with a smaller margin.

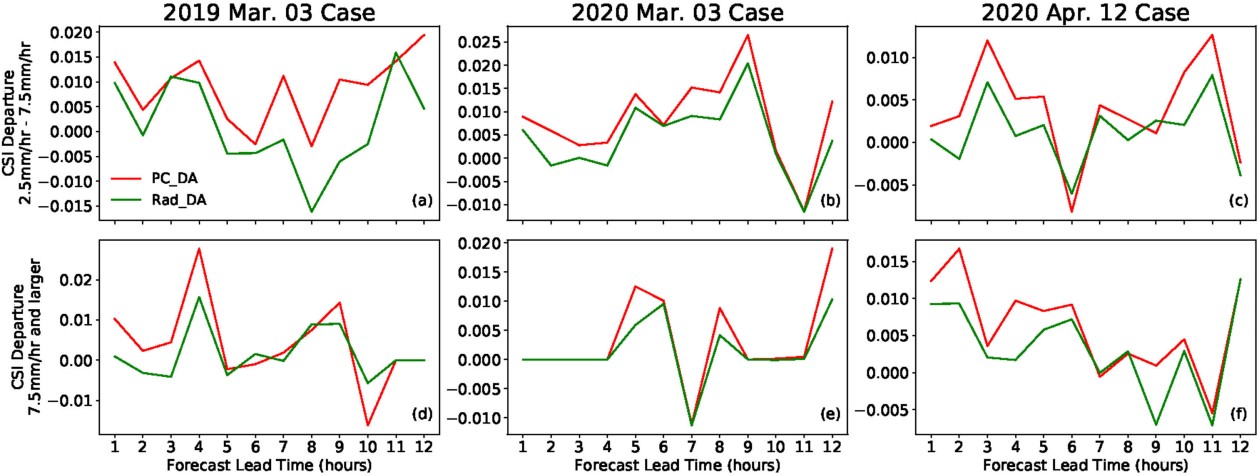

**Figure 5.** CSI departure for precipitation within 2.5 and 7.5 mm/h (**a–c**) and above 7.5 mm/h (**d–f**) from PC-score assimilation (red line) and selected-channel radiance assimilation (green line).

The fixed-layer significant tornado parameter ($STP$), derived from Equation (5), is closely related to tornado outbreaks with F2 intensity or higher [41] when the value is higher than 1. The algorithm considers the 0–6 km bulk wind difference ($BWD_{6km}$), 0–1 km storm-relative helicity ($SRH_{1km}$), surface-based convective available potential energy ($sbCAPE$), condensation-level height ($LCL$), and surface-parcel convective inhibition ($sbCIN$). In this part, the grid-point STP derived from the forecast output at the tornado outbreak time (rounded to the hour) was interpolated to each observed tornado outbreak location using the bi-linear interpolation method. Figure 6 displays the STP intensity difference between the PC-score and selected-channel radiance assimilation experiment with the baseline STP deducted from each experiment. In all three cases, the STP from the PC-score assimilation experiment is higher than that from selected-channel radiance assimilation. This phenomenon demonstrates that assimilating PC scores extracted from full-spectrum hyperspectral IR sounding observations can boost the forecast STP intensity at the tornado outbreak location, which could help increase the probability of detection (POD) for tornadoes.

$$STP_{fixed\ layer} = \frac{sbCAPE}{1500J/kg} \cdot \frac{2000m - LCL}{1000m} \cdot \frac{SRH_{1km}}{150m^2/s^2} \cdot \frac{BWD_{6km}}{20m/s} \cdot \frac{200J/kg + sbCIN}{150J/kg} \quad (5)$$

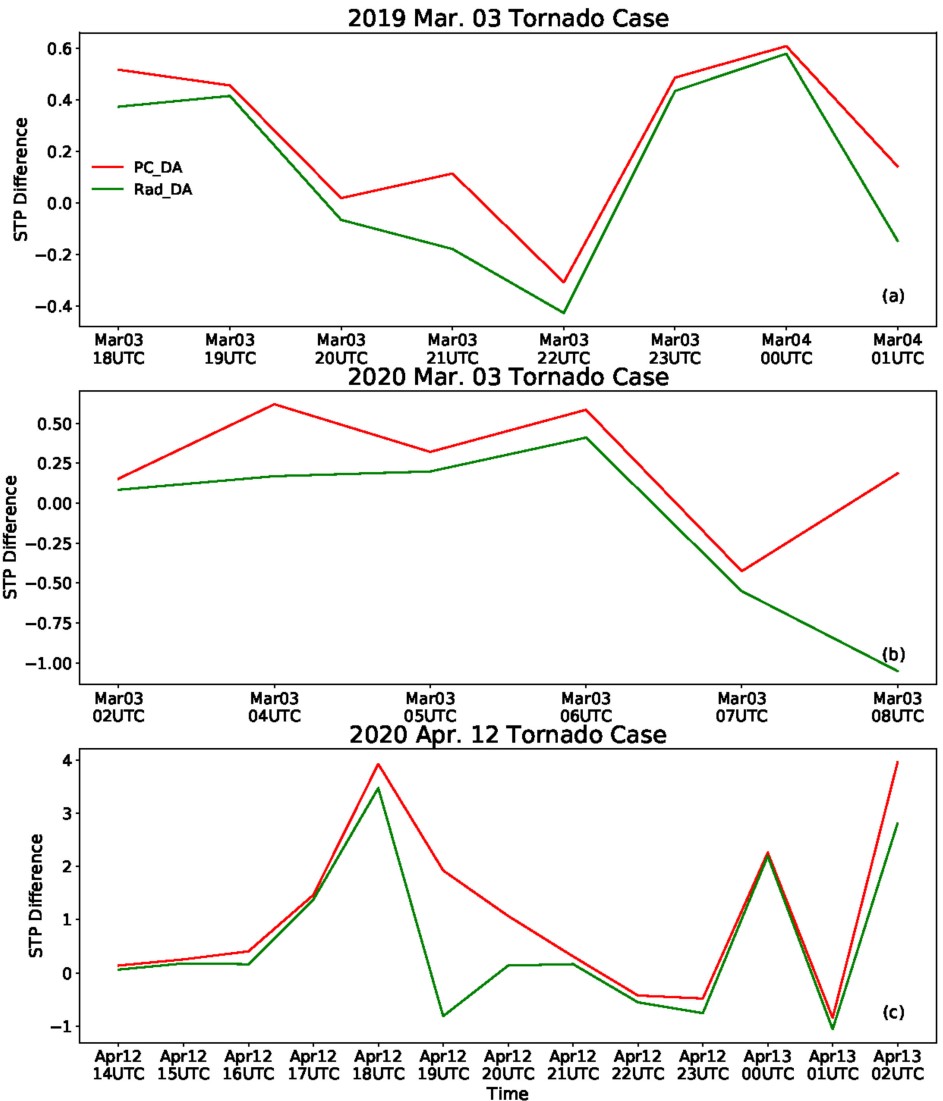

**Figure 6.** PC-score assimilation (red line) and selected-channel radiance assimilation (green line) STP departure for the 2019Mar03 (**a**), 2020Mar03 (**b**), and 2020Apr12 (**c**) cases.

### 3.2. PC-Score vs. Selected-Channel Radiance Assimilation: Four-Month-Long Evaluation

The results presented in Section 3.1 demonstrate that assimilating full-channel IR radiance–derived PC scores can enhance the NWP accuracy for tornado and precipitation prediction compared to selected-channel radiance assimilation. However, this conclusion is derived solely from case studies and requires further long time series validation. Slightly different from the system settings in Section 3.1, this quasi-operational system in this section generates an 18 h lead-time forecast at 00:00 and 12:00 UTC every day. The evaluation in this section comprises a comparison of the PC-score DA system's forecast against channel-selection DA (CTL) output from 31 August 2021 to 1 January 2022.

The initial condition's RMSE profiles (Figure 7a–d) demonstrate that the initial condition generated from PC-score assimilation agrees better than CTL to aircraft and radiosonde observations. Similar to the results in Section 3.1, the improvements are more detectable at levels above 600 hPa (within 850 and 600 hPa) for temperature (specific humidity). This could result from the fact that (1) the temperature and water vapor's Jacobians (weighting functions) for IASI and CrIS in the PC space above 600 hPa are more sensitive than those below 600 hPa, and (2) the Jacobian's peak levels in the selected-channel experiment are mostly located below 600 hPa (especially for temperature), which reduces the observation's impact on upper level atmospheric field modification. Likewise, the wind field accuracy

can be improved by assimilating hyperspectral IR observation in PC space, as the u- and v-component wind RMSE decreased in the initial condition, regardless of the marginal improvement at 1000 hPa. In the 12 h lead-time forecast evaluation, the temperature prediction ability above 600 hPa in the PC-score assimilation system's performance is still superior. Attributable to the assimilated information content conveyed by PC scores through statistical adjustment via the B matrix in the DA process and the forecast model's dynamical adjustment, the wind prediction above 800 hPa (Figure 7e,g,h) is better than that of the CTL forecast. As for specific humidity, the quasi-operational system's performance supersedes that of CTL from 1000 to 300 hPa (Figure 7d). The anomaly correlation coefficient (ACC) can represent the systematic bias of a forecast by calculating the spatial correlation between the forecast and analysis anomalies. The average ACC of temperature and specific humidity (dots in Figure 8a,b) reveals that using the initial condition from PC-score assimilation can alleviate the forecast bias. The deviation in ACC between the minimum (left-hand error bar) and maximum (right-hand error bar) demonstrates that the forecast results from the PC-score assimilation system exhibit consistently higher performance compared to those of CTL. The middle- and upper-level wind evaluation results agree with the conclusions derived from the temperature and water vapor evaluation, although the low-level evaluation comes out in favor of CTL.

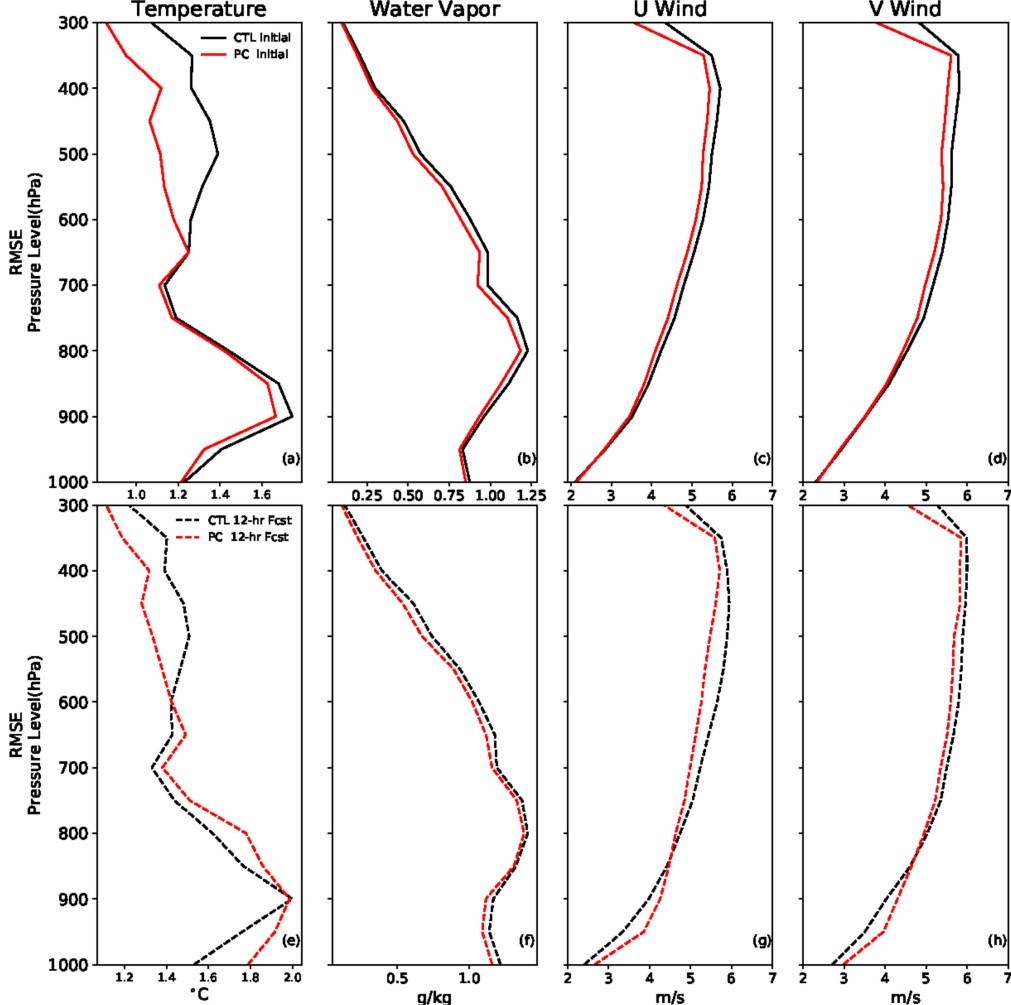

**Figure 7.** The temperature (**a**,**e**), specific humidity (**b**,**f**), u-component wind (**c**,**g**), and v-component (**d**,**h**) wind RMSE profiles from the PC-score assimilation experiment and CTL analysis (**a**–**d**) and 12 h lead-time forecast (**e**–**h**).

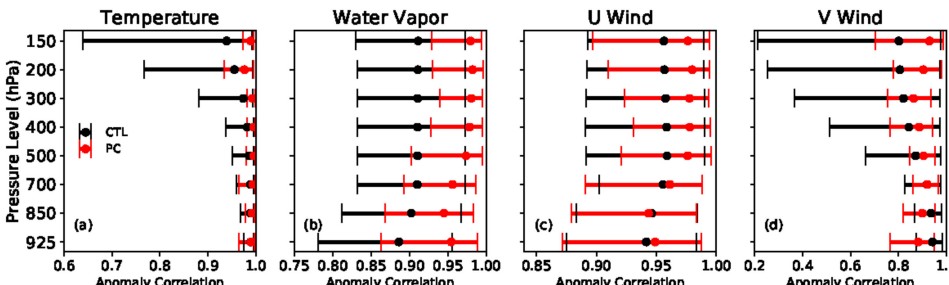

**Figure 8.** The ACC profiles of temperature (**a**), specific humidity (**b**), u-component wind (**c**), and v-component wind (**d**) from the PC-score assimilation system (red) and CTL (black).

In the precipitation forecast evaluation, we first focused on the performance difference in distinguishing the precipitation zone (intensity $\geq$ 1 mm/h) from the non-precipitation area zone (intensity < 1 mm/h) using the Hanssen and Kuipers discriminant (PSS) [42] by evaluating the model precipitation against the NOAA Stage IV quantitative precipitation estimation. The following evaluations exclude the precipitation forecast within a 2 h lead-time due to the forecast model's spin-up effect. As shown in Figure 9a, the PC-score assimilation forecast performs better in distinguishing the precipitation regions, as evidenced by its PSS exceeding that of CTL by 0.11 on average. To evaluate the accuracy in predicting the precipitation intensity, the grid-point rainfall was categorized into light (1–2.5 mm/h), moderate (2.5–7.5 mm/h), and heavy ($\geq$7.5 mm/h) precipitation. The multi-category PSS in Figure 9b depicts that the forecast from the PC-score assimilation prediction system is better at predicting precipitation category than CTL. The Kling–Gupta efficiency (*KGE*), Equation (6) [43], where $r$ is the spatial correlation between observation and simulation $\sigma_{obs}$ ($\sigma_{sim}$), is the standard deviation in observation (simulation). $\mu_{obs}$ ($\mu_{sim}$) is the observation (simulation) mean, demonstrating that the heavy precipitation accuracy from CTL is higher in the first 7 h of the forecast lead-time (Figure 9c) but gets surpassed by the quasi-operational system after 8 h. This improved performance of CTL for heavy precipitation prediction may result from radar reflectivity in its DA system.

$$KGE = 1 - \sqrt{(r-1)^2 + \left(\frac{\sigma_{sim}}{\sigma_{obs}} - 1\right)^2 + \left(\frac{\mu_{sim}}{\mu_{obs}} - 1\right)^2} \qquad (6)$$

In the tornado forecast evaluation, the STP time series (Figure 10a) shows that the quasi-operational system produces a higher intensity than CTL at the outbreak locations. Nevertheless, the POD from the PC-score assimilation-prediction system is not continuously superior: the CTL forecast has a marginal prediction advantage after 9 h of forecast lead-time (Figure 10b). In the radar plot (Figure 11), continuous PC-score assimilation only affects two out of five parameters that comprise the STP, namely $SRH_{1km}$ and $BWD_{6km}$, which are dynamic parameters related to atmospheric motion. In contrast, the $LCL$, $sbCIN$, and $sbCAPE$ are seldom enhanced. To magnify the contribution of the dynamic parameter in the tornado prediction, we substituted the STP with the energy–helicity index: $EHI = SRH_{1km} \cdot BWD_{6km}/16,000$ and re-conduct the evaluation. This time, both the intensity (Figure 12a) and POD (Figure 12b) exceeded those of CTL. In summary, assimilating hyperspectral IR sounding–derived PC scores can enhance the performance of the NWP system in predicting tornado outbreaks.

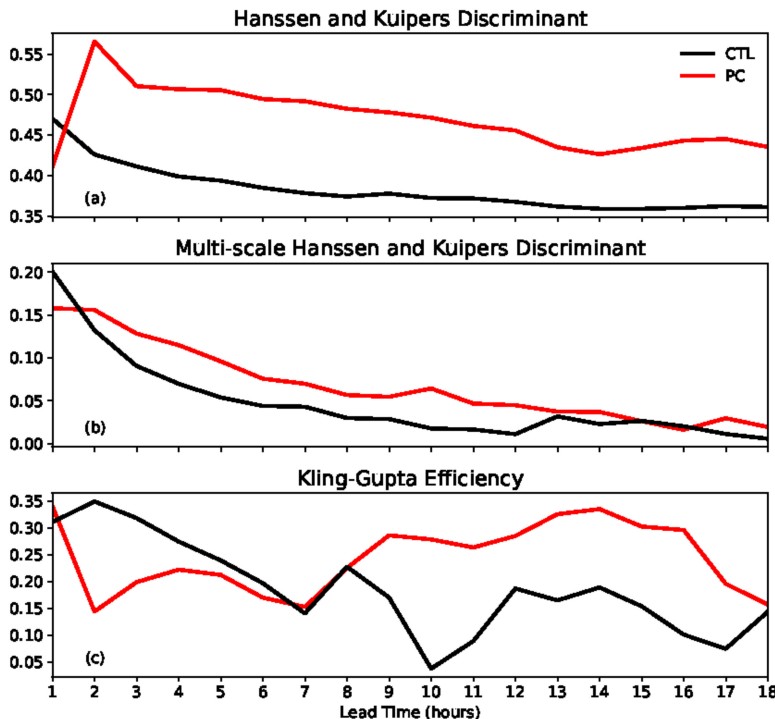

**Figure 9.** Hanssen and Kuipers discriminant (**a**), multi-category Hanssen and Kuipers discriminant (**b**), and Kling–Gupta efficiency (**c**) time series from the PC-score assimilation system (red) and CTL (black).

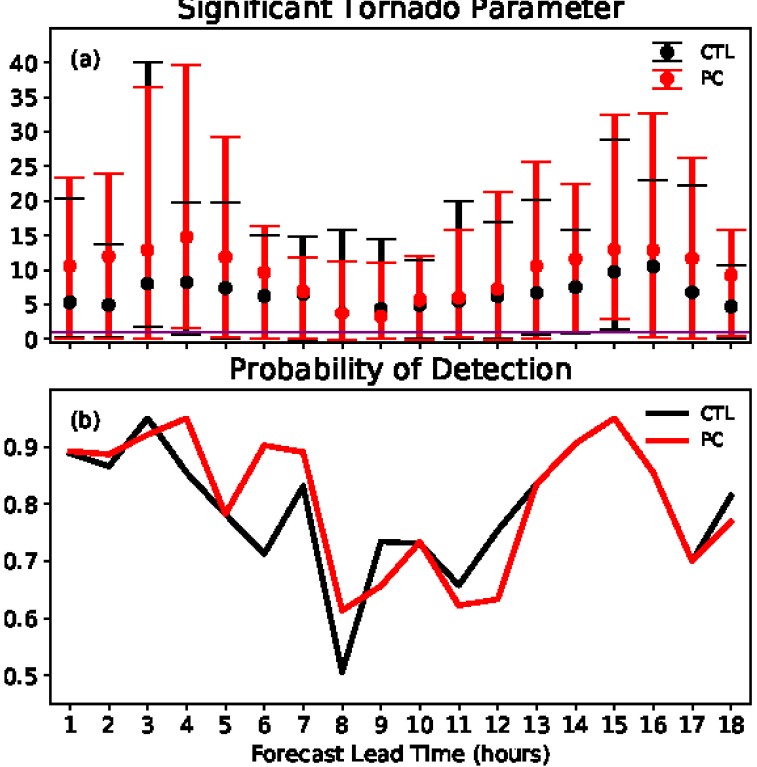

**Figure 10.** Time series of the fixed-layer significant tornado parameter (**a**) and its POD (**b**) derived from the PC-score assimilation system forecast result (red) and CTL (black).

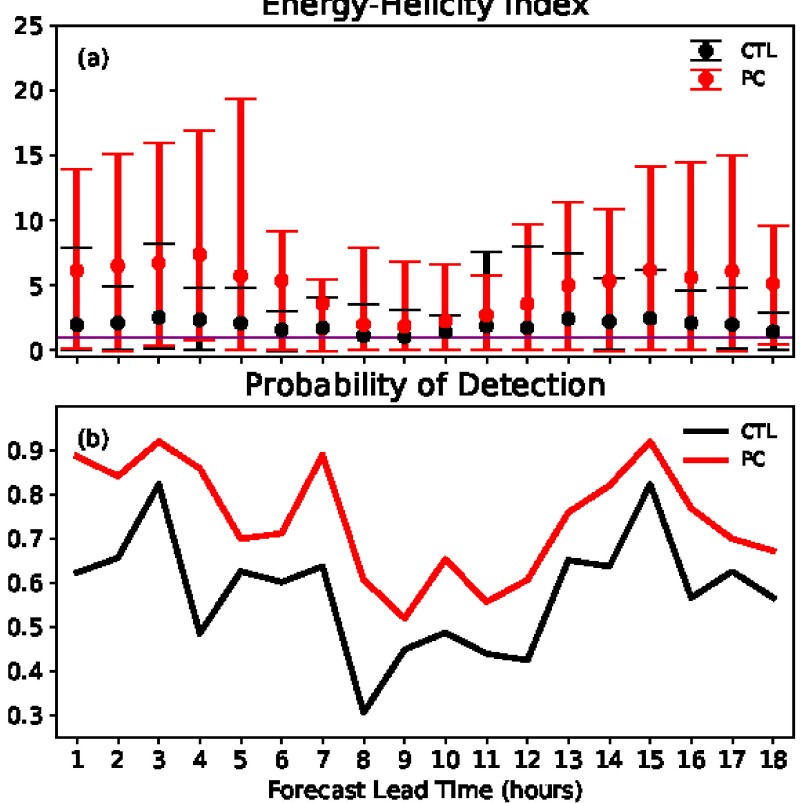

**Figure 11.** The contribution of each variable in the calculation of the significant tornado parameter, with the red (black) line representing the PC-score experiment forecast result (CTL).

**Figure 12.** Time series of the EHI (**a**) and its POD (**b**) derived from the PC-score assimilation experiment forecast result (red) and CTL (black).

### 3.3. Convection-Resolving Resolution Case Studies

As a preliminary examination of the PC-score assimilation system's compatibility to different NWP models and different horizontal resolution, we replaced the WRF with UFS-SRW in the system and increased the horizontal resolution to 3 km. The new domain and physical scheme settings follow the default Rapid Refresh Forecast System (RRFS) description, which is embedded in UFS-SRW version 1.0.0 [44]. Two case studies of tornado and hail outbreaks illustrate that the PC-score assimilation can retain a positive impact for severe weather prediction at a convection-resolving scale, regardless of the difference between the IFOV resolution (lower than 12 km) and the initial condition's horizontal resolution.

Case 1 is a series of hailstorms that occurred in Wisconsin and Michigan between 12:00 UTC 7 September 2021 and 12:00 UTC 8 September 2021. The consistency between the spatial distribution of the hourly maximum significant hail parameter (SHiP) is as follows:

$$SHiP = \frac{MUCAPE \cdot MUMixingRatio \cdot LapseRate_{700-500mb} \cdot BWD_{0-6km} \cdot 500mbTemperature}{42,000,000} \tag{7}$$

where $MUCAPE$ is the most unstable convective available potential energy, $MUMixingRatio$ is the water vapor mixing ratio at the most unstable level, $LapseRate_{700-500mb}$ is the lapse rate from 700 to 500 mb, and $500mbTemperature$ is the ambient air temperature at 500 mb. The hail outbreak location shows that the forecast initialized at 00:00 UTC 7 September in the PC-score assimilation experiment captures the hail outbreaks in areas I, II, and III (Figure 13b), while the SHiP in the NOAA High-Resolution Rapid Refresh (HRRR) forecast indicates the potential for a hail outbreak over these three areas is low (Figure 13a). The HRRR forecast initialized at 12:00 UTC 7 September misses many hail outbreaks in area IV (Figure 13c), but rarely can we find any misses in the PC-score assimilation's forecast over that area (Figure 13d). These results demonstrate that the PC-score assimilation experiment's POD is higher than that of the HRRR forecast.

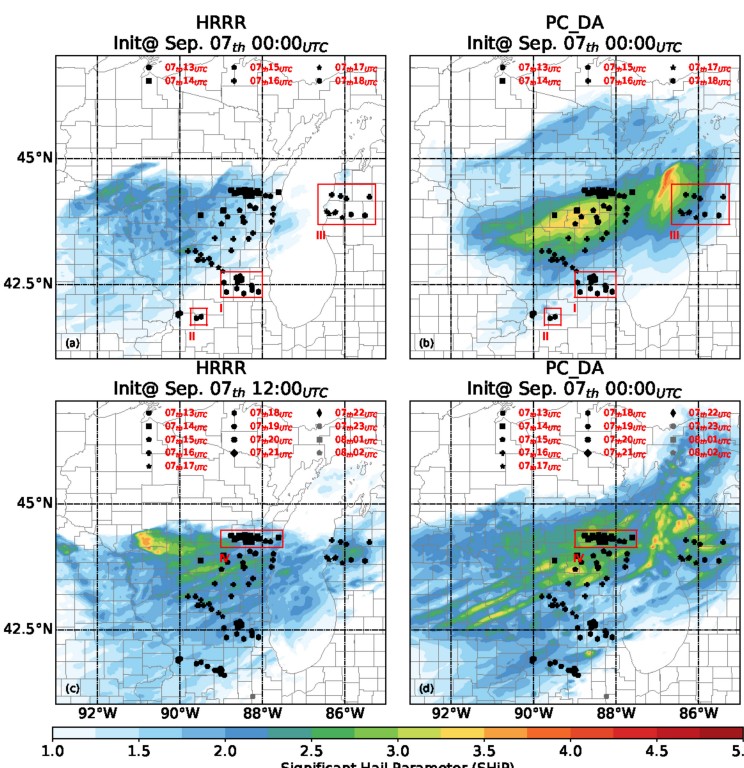

**Figure 13.** Comparison between the HRRR and pseudo-operational forecasts initialized at 00:00 UTC 7 September (**a**,**b**) and 00:00 UTC 7 September (**c**,**d**). The black and gray marks (with different shapes representing the outbreak time) are the hail outbreak locations.

From 12:00 UTC 15 December to 12:00 UTC 16 December, multiple tornadoes occurred across Nebraska, Iowa, Minnesota, and Wisconsin (Case 2). In the forecast initialized at 12:00 UTC 15 December, the PC-score assimilation experiment (Figure 14b) has a smaller miss ratio than HRRR (Figure 14a) because of the higher agreement between the EHI and tornado outbreaks in Nebraska (area I). The same results are found in the forecast initialized at 00:00 UTC 16 December over area II (Figure 14c,d).

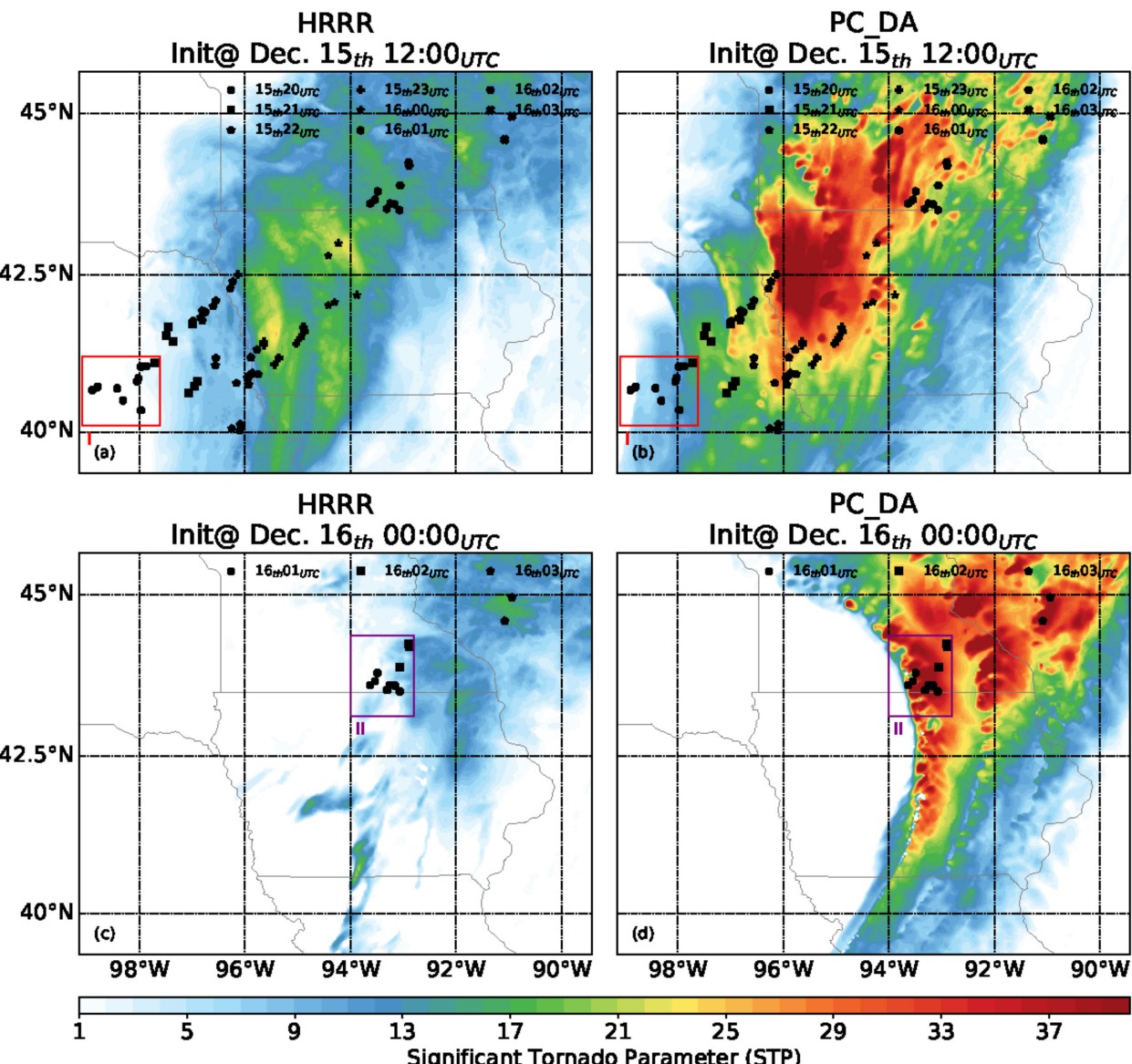

**Figure 14.** Comparison between the HRRR and pseudo-operational forecasts initialized at 12:00 UTC 15 December (**a**,**b**) and 00:00 UTC 16 December 00:00 UTC (**c**,**d**). The black and gray marks (with the different shapes representing the outbreak time) are the hail outbreak locations.

## 4. Discussion

In Sections 3.1 and 3.2, PC-score assimilation reveals its potential to improve the NWP system's forecast accuracy for temperature, water vapor, light to moderate precipitation, and the mid- to upper-level wind field. However, the higher standard deviation in u- and v-component wind as well as the lower performance in heavy precipitation prediction over the first 8 h lead-time compared to the selected-channel assimilation experiment reminds us that the PC-score assimilation scheme still needs improvement before operational deployment. PC-score assimilation's compatibility with the Hyperspectral Infrared

Atmospheric Sounder (HIRAS) onboard FY-3D/E and Infrared Fourier Spectrometer 2 onboard Meteor-3M also needs further investigation.

The two case studies with higher resolution (3 km) using UFS-SRW as a forecast module indicate that PC-score assimilation in Section 3.3 can retain its performance at a different resolution since the performance demonstrated in the experiments is mostly comparable to HRRR. The additional performance improvement can be attributed to (1) the hybrid data assimilation system's improved performance at a convection-resolving resolution [45,46]; (2) the forecast module's coherence of the dynamical core, the Finite-Volume Cubed-Sphere (FV3) dynamical core, in the regional and global ensemble prediction scenario. However, additional experiments are still needed for precise confirmation.

While the performance of the PC-score assimilation is solid, it still has limitations. First, the PC-based fast radiative-transfer model can only calculate radiance or brightness temperature in the IR region, which significantly constrains our DA system from assimilating other atmospheric observations. In other words, these DA systems must include additional forward operators, including but not limited to radiative-transfer models, to assimilate a comprehensive observation dataset, e.g., passive/active microwave sounding and Global Positioning System (GPS) Radio Occultation observations. Thus, investigating how well the PC-score assimilation scheme can perform in an integrated DA environment is one of our priorities in the future. Second, it has become a general conclusion that the assimilation of a geostationary hyperspectral infrared sounder has a positive impact in a regional NWP system [47,48]. Whether PC-score assimilation generates equivalent impact when assimilating the geostationary hyperspectral IR sound information still needs investigation. Third, a practicable approach of combining geostationary and low-earth-orbiting hyperspectral IR observations in the DA process also needs to be developed.

## 5. Conclusions

This study demonstrates that assimilating PC scores derived from hyperspectral IR sounding observations via the hybrid 3DVar scheme can benefit the NWP system, and improvements can be detected if compared against the selected-channel radiance assimilation. The results presented in this study exhibit noticeable bias reduction in the initial condition and improvements in several forecast variables, such as precipitation and tornado outbreaks. We tested the PC-score assimilation scheme without assimilating other types of observations, which can highlight the influence of hyperspectral IR observations on improving the NWP performance. Case studies and a 4-month-long experiment suggest that PC-score assimilation can produce a more reliable and trustworthy initial condition and forecast, except for the low-level temperature and wind fields. To summarize, the direct assimilation of PC scores demonstrates potential and the capability to improve the accuracy of NWP relative to radiance assimilation by correcting the initial condition with abundant and independent information from hyperspectral IR observations conveyed by PC scores.

**Author Contributions:** Conceptualization, Q.Z.; methodology, Q.Z.; software, Q.Z.; validation, Q.Z.; data retrieving, M.S.; writing—original draft preparation, Q.Z.; writing—review and editing, M.S.; funding acquisition, M.S.; All authors have read and agreed to the published version of the manuscript.

**Funding:** The research is funded by Special Science and Technology Innovation Program for Carbon Peak and Carbon Neutralization of Jiangsu Province (Grant Number BE2022612).

**Institutional Review Board Statement:** Not applicable.

**Informed Consent Statement:** Not applicable.

**Data Availability Statement:** The online access URLs are provided at the first location where the data are mentioned in the article.

**Acknowledgments:** We thank editors for their academic editing services. We would also like to show our gratitude to all "anonymous" reviewers for their insights and comments on an earlier version of the manuscript, although any errors are our own and should not tarnish the reputations of these esteemed persons.

**Conflicts of Interest:** The authors declare no conflict of interest.

## Appendix A

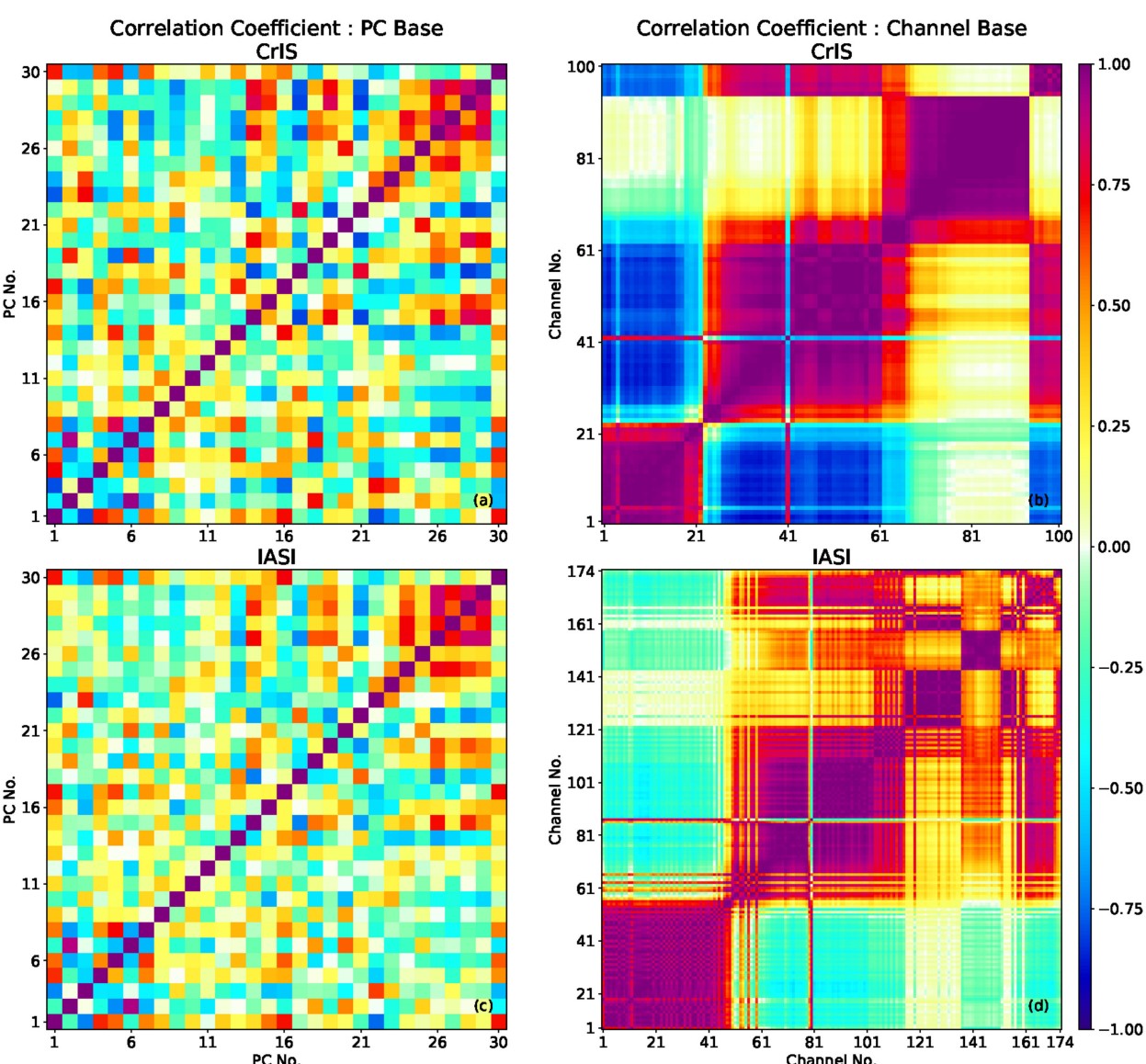

**Figure A1.** Correlation coefficient matrix in principle component (**a**,**c**) and radiance (**b**,**d**) space for CrIS (**a**,**b**) and IASI (**c**,**d**).

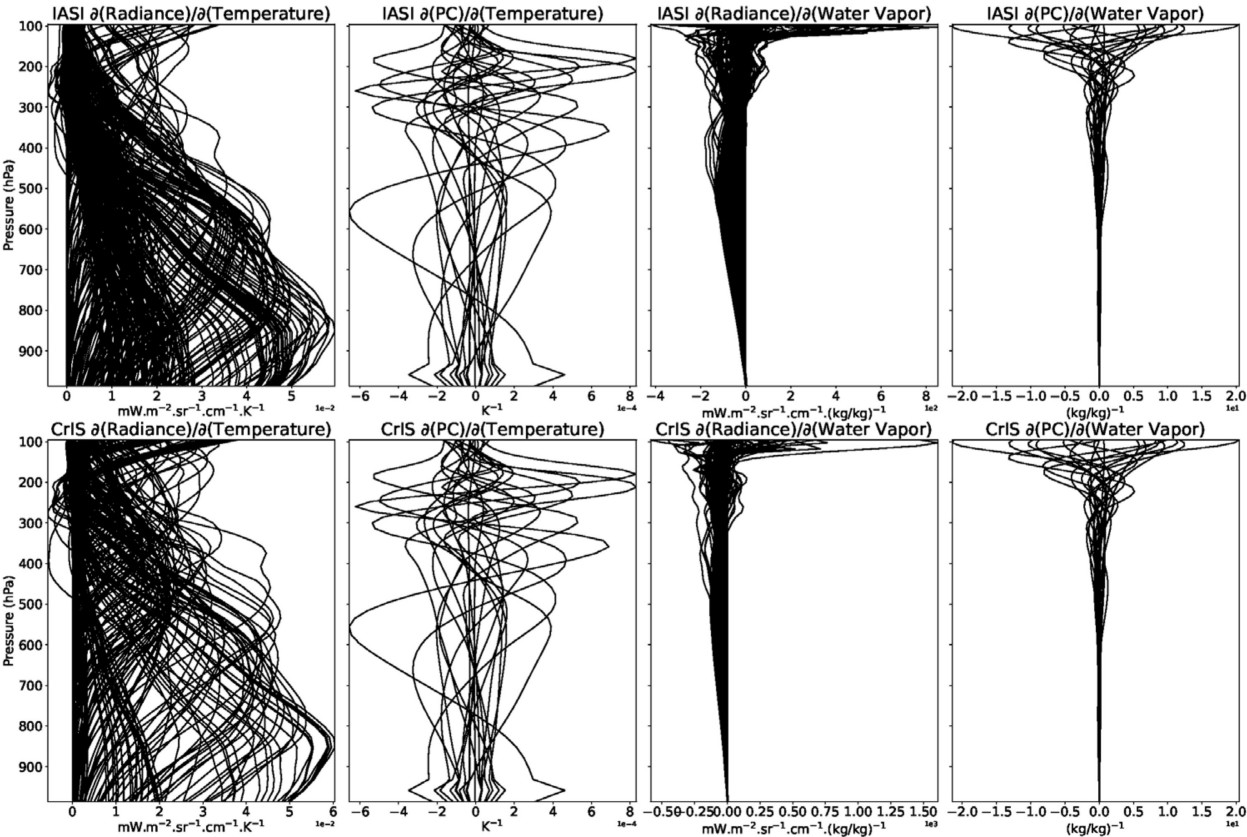

**Figure A2.** Weighting function for CrIS and IASI in PC and radiance space. Profile dataset comes from European Organization for the Exploitation of Meteorological Satellites (EUMETSAT) 60-level sample profile dataset from the Monitoring Atmospheric Composition and Climate (MACC) project (available at https://nwp-saf.eumetsat.int/site/download/profile_datasets/60l_macc.dat.tar.bz2 (accessed on 3 February 2023)).

**Table A1.** Contingency table.

| | | Contingency Table | |
|---|---|---|---|
| | | **Observation** | |
| | | Happen | Not Happen |
| | Happen | a | b |
| Forecast | Not Happen | c | d |

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
