# Peer review of "Impact of Hyperspectral Infrared Sounding Observation and Principal-Component-Score Assimilation on the Accuracy of High-Impact Weather Prediction"

_atmosphere, doi:10.3390/atmos14030580_

Round 1

Reviewer 1 Report

The paper presents an interesting approach for the assimilation of hyperspectral infrared sounding in weather forecast.

As general comment, the content of the paper must be improved with a better and more exhaustive explanation of the adopted procedure expanding the session Materials and Methods. The paper presents a lot of results and graphics but the section devoted to Discussion is scarce. The same comment is valid for Conclusions.

Few notes in detaili

-        Equation 4 at pag. 6 is wrong, please check it OBS Amount should be (OBS amount )2 .

-        Equation 4 at page. 6 definition of ??? ?????? is missing.

Author Response

Dear Reviewer 1,

Deeply appreciated your time effort reviewing the manuscript and the insightful input are of vital importance to us.

Here are our point-to-point replies:

Point 1 As general comment, the content of the paper must be improved with a better and more exhaustive explanation of the adopted procedure expanding the session Materials and Methods. The paper presents a lot of results and graphics but the section devoted to Discussion is scarce. The same comment is valid for Conclusions.

Re: The cloud-screening and quality-control approach are added to the “Materials and Methods” section (line 134 - 151). We also re-write the “Discussion” and “Conclusion” (line 369 - 414).

Point 2 Equation 4 at pag. 6 is wrong, please check it OBS Amount should be (OBS amount )2

Re: Corrected.

Point 3 

Equation 4 at page. 6 definition of OBS Amount is missing.

Re: Added (line 183).

Best regards,

All Authors

Reviewer 2 Report

Review of “Impact of Hyperspectral Infrared Sounding Observation Principal Component Score Assimilation on The Accuracy of High Impact Weather Prediction”, atmosphere-2243028

General comments:

This study introduced a principal-component-based data assimilation (DA) scheme for the Hyperspectral Infrared Sounding observations onboard IASI and CrIS and evaluated their DA impact on the analysis and prediction of high-impact weather events. This is a valuable study in the area of satellite DA and provides a potentially efficient approach to assimilate multi-channel sounding observations. The manuscript is logically organized and well written. I would recommend a minor revision.

Minor comments:

1.      Lns 109-110, why do the authors use the weight 80% for the static B while 20% for the ensemble-based B? Some studies have demonstrated a nearly full ensemble-based B may be used for meso-scale DA like hurricanes because of its rapid evolution and large spatial gradient (e.g., Feng and Wang, 2021, Lu and Wang, 2019). Do the authors use this setup for both the DA experiments at 12 km and 3 km?

2.      Fig. 1, The decomposed principal components (PC) should be independent or orthogonal. Why do they show covariance across PCs in the left panel?

3.      Research and operational applications of the Geostationary Interferometric Infrared Sounder (GIIRS) observations onboard China’s FengYun-4A geostationary satellite have also been implemented in the DA and prediction with direct (e.g., Yin et al. 2021) and indirect (e.g., Feng et al. 2022) assimilation schemes. It may be helpful to mention some of these studies in the introduction.

4.      Lns 226-227, “the improvements are more detectable at levels above 600hPa”, how to explain this?

5.      In section 3.3, the authors compared their results to the HRRR system. What observations are assimilated in the HRRR system? Are there Infrared Sounding observations assimilated for the HRRR system? If not, is it a fair comparison for these two experiments?

6.      In section 3.3, the model resolution increases to 3 km from 12 km. Based on the setup of the DA system, the improved analysis may be attributed to the more reasonable and accurate simulation of the first guess with a 3-km resolution model because either the static or the ensemble-based background covariance does not change. This point may be slightly discussed in the manuscript.

7.      The Summary section is too simplified. Please expand and summarize the main conclusions.

References:

Feng, J. and X. G. Wang, 2021: Impact of increasing horizontal and vertical resolution of the hurricane WRF model on the analysis and prediction of Hurricane Patricia (2015). Mon. Wea. Rev., 149(2), 419–441.

Lu, X., and X. Wang, 2019: Improving hurricane analyses and predictions with TCI, IFEX field campaign observations, andCIMSS AMVs using the advanced hybrid data assimilation system for HWRF. Part I: What is missing to capture the rapid intensification of Hurricane Patricia (2015). Mon. Wea. Rev.,147, 1351–1373

Yin, R., Han, W., Gao, Z., & Li, J. (2021). Impact of High Temporal Resolution FY-4A Geostationary Interferometric Infrared Sounder (GIIRS) Radiance Measurements on Typhoon Forecasts: Maria (2018) Case With GRAPES Global 4D-Var Assimilation System. Geophysical Research Letters48(15).

Feng, J., X. Qin, C. Wu, P. Zhang, L. Yang, X. S. Shen, W. Han, Y. Z. Liu, 2022: Improving typhoon predictions by assimilating the retrieval of atmospheric temperature profiles from the FengYun-4A's Geostationary Interferometric Infrared Sounder (GIIRS), Atmospheric Research, 280, 106391, ISSN 0169-8095.

Grammatical errors:

1.      Ln 30, changed to “The use of hyperspectral …”

Author Response

Dear Reviewer 2,

Deeply appreciated your time effort reviewing the manuscript and the insightful input are of vital importance to us.

Here are our point-to-point replies:

Point 1 Lns 109-110, why do the authors use the weight 80% for the static B while 20% for the ensemble-based B? Some studies have demonstrated a nearly full ensemble-based B may be used for meso-scale DA like hurricanes because of its rapid evolution and large spatial gradient (e.g., Feng and Wang, 2021, Lu and Wang, 2019). Do the authors use this setup for both the DA experiments at 12 km and 3 km?

Re: The references are modified base on the suggestions (line 383). The reasons for selecting 80% static and 20% ensemble B matrix can be as follows: (1) the GEFS ensemble forecast product with public accessibility only has 30 ensemble members and its vertical resolution is super-rendered (interpolated from model level to pressure level), a dynamical down-scaling using GEFS as initial condition can generate the spatial variability but also brings unpredictable bias spatially and vertically which cannot be resolved unless we use more ensemble products with equivalent or higher vertical resolution to generate the ensemble members. (2) Using less ensemble members can save computational resources, and with these extra resources we can perform a longer (e.g. 4 month in this study) but more operational-like (Research to Operation, a.k.a, R2O) experiment; To partially overcome the disadvantages, we added two high-spatial resolution (3km) case studies in section 3.3 using the same methodology.

Point 2 Fig. 1, The decomposed principal components (PC) should be independent or orthogonal. Why do they show covariance across PCs in the left panel?

Re: The cross-PC covariance (off-diagonal value in PC’s R Matrix) can not be zero are attributed to: The decomposition of observed radiances highly relies on the Empirical Orthogonal Functions (EOFs). These EOFs are also called coefficients in PC-based radiative transfer models and calculated from the LBLRTM-simulated instrumental radiances. In HTFRTC, only a few trace gases are considered (O3, CO2, CH4) in the LBLRTM simulation when generating the EOFs to make sure that these EOFs are suitable for temperature and water vapor inversion while ignoring the impact from other trace gases, e.g. NO, NO2, N2O. It could be the incomplete consideration of trace gases that causes this issue. Additional arguments, the off-diagonal value are almost zero in the first 15 PCs and becomes detectable afterwards, which could be a clue that “all the off-diagonal value can be zero if all trace gases are properly considered in coefficient (EOFs) generation”.

Point 3 Research and operational applications of the Geostationary Interferometric Infrared Sounder (GIIRS) observations onboard China’s FengYun-4A geostationary satellite have also been implemented in the DA and prediction with direct (e.g., Yin et al. 2021) and indirect (e.g., Feng et al. 2022) assimilation schemes. It may be helpful to mention some of these studies in the introduction.

Re: Added to line 397.

Point 4 Lns 226-227, “the improvements are more detectable at levels above 600hPa”, how to explain this?

Re: The explanation “This could result from the fact that (1) the temperature and water vpaor’s Jacobians (weighting functions) for IASI and CrIS in PC space above 600hPa are more sensitive than those below 600hPa and (2) the Jacobian’s peak levels in selected-channel experiment are mostly located below 600hPa (especially for temperature), which reduces the observation’s impact on upper level atmospheric field modification” is added to line 250 - 254.

Point 5 In section 3.3, the authors compared their results to the HRRR system. What observations are assimilated in the HRRR system? Are there Infrared Sounding observations assimilated for the HRRR system? If not, is it a fair comparison for these two experiments?

Re: Starting From version 4, HRRR assimilates hyperspectral IR sounding observations (https://rapidrefresh.noaa.gov/pdf/Alexander_AMS_NWP_2020.pdf) along side with other observations, to this point, it is absolutely not an equal comparison. However, the scope of section 3.3 is preliminarily demonstrating PC score assimilation scheme can retain the performance at a different horizontal resolution in a different mode. Therefore, the result can be considered as reasonable if the experiment does not have a significant performance downgrade compared to HRRR.

Point 6 In section 3.3, the model resolution increases to 3 km from 12 km. Based on the setup of the DA system, the improved analysis may be attributed to the more reasonable and accurate simulation of the first guess with a 3-km resolution model because either the static or the ensemble-based background covariance does not change. This point may be slightly discussed in the manuscript.

Re: Added to the discussion section (line 372 - 379).

Point 7 The Summary section is too simplified. Please expand and summarize the main conclusions.

Re: We re-write the “Disscussion” and “Conclusion” section (line 369 - 414).

Point 8 Ln 30, changed to “The use of hyperspectral …”

Re: Changed.

Best regards,

All Authors

Reviewer 3 Report

The paper is about how  the hyperspectral infrared sounding principal component score assimilation affects the accuracy of weather prediction. The paper is well written.  However, I suggest to address the main question by the research clearly in the introduction section  and answer it in the conclusion as well, to make the goal more clear. Overall, I would recommend  the paper fot publication. Minor points are as follows:

-          Section 2: It should be mentioned that what are the initial and lateral boundary conditions on the WRF performance?

-          Sec. 2: How long is the spin up time for the WRF-runs?

-          Table 1: What is the reason to choose this configuration? Is there any reference that previously suggest this configuration over the region?

-          Fig 4: in the title RMSD should be changed to  RMSE.

-          Fig. 4: What is the interpretation for having a higher RMSE departure for specific humidity in PC-score assimilation near the surface?

-          It is suggested to introduce the efficiency methods (in fig.9) in the methodology section.

Author Response

Dear Reviewer 3,

Deeply appreciated your time effort reviewing the manuscript and the insightful input are of vital importance to us.

Here are our point-to-point replies:

Point 1 Section 2: It should be mentioned that what are the initial and lateral boundary conditions on the WRF performance?

Re: The sentence “In the following experiments, the initial and boundary condition are provided by RAP forecast product.” is added to manuscript (line 158).

Point 2 Sec. 2: How long is the spin up time for the WRF-runs?

Re: There are 12 analysis cycles before the forecast cycle, and they are warm-start runs. Consequently, no spin-up time is needed in the forecast cycle. This design is in accordance with the RAP system.

Point 3 Table 1: What is the reason to choose this configuration? Is there any reference that previously suggest this configuration over the region?

Re: The configuration is chosen based on the NOAA/NCEP RAP system’s forecast model settings (reference # 36). Also we highlight the this reference at line 157.

Point 4 Fig 4: in the title RMSD should be changed to RMSE.

Re: Changed.

Point 5 Fig. 4: What is the interpretation for having a higher RMSE departure for specific humidity in PC-score assimilation near the surface?

Re: This situation could be related to the uncertainty of surface emissivity over land. Unlike the water surface, the surface emissivity over land is highly dependent on land-use categories. A 13km horizontal resolution can represent large-scale land use type but may not be able to reveal the variability where there are multiple land type within a grid point.

Point 6 it is suggested to introduce the efficiency methods (in fig.9) in the methodology section.

Re: Equation added at line 298.

Best regards,

All Authors

Round 2

Reviewer 2 Report

The authors have addressed all my comments. I would recommend acceptance of the manuscript in its present form.